# Costs Attributable to Falls Based on Diagnosis-Related Groups (DRGs) Analysis of Hospitalised Patients: A Case–Control Study

**DOI:** 10.3390/nursrep15090323

**Published:** 2025-09-05

**Authors:** Mercedes Fernández-Castro, Noel Rivas-González, Belén Martín-Gil, Pedro Luis Muñoz-Rubio, Rocío Lozano-Pérez, Pilar Rodríguez-Soberado, Marife Muñoz

**Affiliations:** 1Research Support Unit, Valladolid University Clinical Hospital, 47005 Valladolid, Spain; mefernandezc@saludcastillayleon.es (M.F.-C.); mfmunozm@saludcastillayleon.es (M.M.); 2Valladolid Biosanitary Research Institute (IBIOVALL), Valladolid University Clinical Hospital, Calle Rondilla Santa Teresa, 47010 Valladolid, Spain; 3Continuing Education Department, Valladolid University Clinical Hospital, 47005 Valladolid, Spain; 4Department of Nursing Care Information Systems, Valladolid University Clinical Hospital, 47005 Valladolid, Spain; bmartingi@saludcastillayleon.es; 5Neurology Unit, Valladolid University Clinical Hospital, 47005 Valladolid, Spain; plmunoz@saludcastillayleon.es; 6Internal Medicine Unit, Valladolid University Clinical Hospital, 47005 Valladolid, Spain; rlozanop@saludcastillayleon.es; 7Care Service of the Regional Health Management of Castile and León, 47007 Valladolid, Spain; mprodriguezs@saludcastillayleon.es

**Keywords:** Diagnosis-Related Groups (DRG), falls, inpatient falls, nursing management, economic analysis

## Abstract

**Background/objectives**: Falls are the most common adverse events in hospitals. This study aimed to estimate excess hospitalisation costs attributable to inpatient falls, using Diagnosis-Related Group (DRG) relative weights as a proxy for resource consumption. **Methods**: Case–control study. Cases included patients who had sustained a fall during hospitalisation between 2020 and 2022 in 19 inpatient units. Controls were selected with matching technique based on age and admission period. Diagnosis-Related Groups and their resource consumption and cost estimators (relative weights) were provided by the Hospital’s Coding Unit. **Results**: A total of 613 falls were analysed against 623 controls. The Diagnosis-Related Group ‘Lower limb amputation except toes’ was associated with a fourfold higher risk of falling compared to others. Five more were identified in which the case group incurred significantly higher costs than the control group. These included three surgical Diagnosis-Related Group, ‘Urethral and transurethral procedures’, ‘Heart valve procedures without acute myocardial infarction or complex diagnosis’, and ‘Arterial procedures on the lower limb’, and two medical, ‘Heart failure’ and ‘Major pulmonary infections and inflammations’. **Conclusions/Implications for practice**: Identifying Diagnosis-Related Groups in which falls are associated with increased hospitalisation costs allows for a comprehensive assessment of the process, taking into account resource consumption and the clinical characteristics of hospitalised patients. These findings will enable nurses to develop targeted strategies to enhance the safety of hospitalised patients that contribute to the sustainability of the healthcare system.

## 1. Introduction

The study of falls in hospitalised adult patients is an emerging field, as falls are the most common adverse events in hospitals worldwide. They represent a significant issue that undermines the quality of care, prolongs hospital stays, and worsens patient recovery, resulting in increased costs for healthcare systems [1,2]. In Spain, studies estimate an incidence of falls of approximately 2.7% among hospitalised patients over 65 years of age, with a fall rate of 1.61 per 1000 patient-days. However, it is recognised that the actual incidence is higher than that reported in official notification systems [3]. Population ageing, high incidence rates, long-term consequences, and the economic burden of falls are expected to increasingly impact healthcare systems. Over the coming years, both the number of falls and their associated costs are projected to rise substantially, potentially jeopardising the sustainability of healthcare systems [4]. Recent studies have focused on evaluating the effectiveness of hospital-based fall prevention programmes and interventions [5,6,7], as well as analysing the financial burden of falls by quantifying additional procedures, surgical interventions, medications, diagnostic tests, and extended hospital stays resulting from falls. However, studies assessing costs often highlight the challenge of accurately determining the direct financial impact of falls, given the influence of multiple factors. While indirect costs may be more difficult to quantify, they are no less significant [8,9,10,11].

The Diagnosis-Related Group (DRG) system enables the analysis of costs incurred by each patient during hospitalisation. It was originally developed in the United States in 1969 and has since been adopted by numerous countries across five continents [12]. This system classifies hospital episodes based on clinical characteristics and resource consumption, grouping diagnoses using the Clinical Classifications Software Revised (CCSR v2022.1), a classification tool within the Healthcare Cost and Utilization Project (HCUP) in the United States. In Spain, coding is based on the International Classification of Diseases (ICD-10).

The Spanish Ministry of Health periodically compiles the costs of hospitalisation processes through an agreement with the Spanish Hospital Costs Network (RECH in Spanish). This is a network of hospitals that provides the total cost of hospitalisation episodes, systematically collected after the financial year is closed, taking into account DRG categories. Each category is also assigned a relative weight that reflects the average consumption of hospital resources for that diagnostic group in comparison with the average case. These weights are calculated from actual hospital cost data using micro-costing methodologies and analytical accounting systems. The process involves collecting and analysing recent real hospitalisation episodes in order to adjust the weights to changes in clinical practice, technology, and cost structures [13]. The Ministry of Health uses DRGs and their weightings as the foundation of the financing system applied to Spain’s Autonomous Communities. The amount a hospital receives for each discharge is calculated by multiplying the weight of the DRG assigned to the episode by a base rate (price per average case), adjusted for factors such as complexity, hospital type, and management agreements. In this way, DRGs make it possible to link funding to activity and efficiency, fostering resource management and transparency [14]. Each DRG is assigned a relative weight, which represents the ratio between the estimated cost of that DRG and the average cost. A relative weight of 1 corresponds to the standard average cost for hospitalised patients within a given DRG. A value above 1 indicates that the cost of that group exceeds the average patient cost. The DRG system assigns a relative weight based on stratified cost levels according to the severity of the hospitalisation episode: minor (Weight = 1), moderate (Weight = 2), major (Weight = 3), and extreme (Weight = 4). These classifications consider patient characteristics, secondary diagnoses, and the procedures performed [14].

Some studies have explored the impact of nursing teams on patient safety and fall prevention in hospitals, highlighting the importance of appropriate nurse staffing levels and organisational structure [15,16]. Some authors have suggested that nursing units with robust safety climates, adequate staffing levels, and high quality care standards are associated with lower rates of falls [17]. However, the Diagnosis-Related Groups (DRGs) system does not directly account for nursing care costs. As outlined above, it is based on diagnoses, procedures, comorbidities, and length of stay. However, this billing model does not consider the different nursing care costs for patients with varying care needs. Previous studies have reported that nursing care costs can vary considerably even within the same DRG, affecting overall hospital costs. Nevertheless, there are currently no universally recognised billing models that accurately measure the impact and cost of nursing care [18].

Our hospital has recognised the significance of this issue and has implemented the Best Practice Guideline (BPG) Preventing Falls and Reducing Injury from Falls, developed by the Registered Nurses’ Association of Ontario (RNAO^®^) [19]. This initiative follows the ‘Knowledge to Action’ framework within the Best Practice Spotlight Organisation^®^ (BPSO^®^) programme, with implementation led by nurses. This includes designating nurses responsible for promoting adherence to the guideline’s recommendations within each hospital unit and fostering a culture of patient safety among their colleagues [20]. To ensure the sustainability of BPG implementation under nurse-led management and optimise its effectiveness, we aim to analyse Diagnosis-Related Groups (DRGs) and their associated relative weight as an estimator of resource consumption and costs in patients who sustained a fall during hospitalisation compared to a control group of non-fallers, in order to identify the excess costs attributable to falls.

## 2. Methods

### 2.1. Study Design

Case–control study. This study is reported in accordance with the STROBE guidelines for observational research.

### 2.2. Participants and Setting

The study population comprised patients aged 18 years or older who were admitted to 19 inpatient units at the Valladolid University Clinical Hospital between 2020 and 2022. This tertiary care hospital is part of the public healthcare network of Castile and León and serves a population of approximately 240,000 inhabitants. The hospital’s service area is characterised by higher ageing, longevity, and dependency rates compared to the rest of the Spanish population, which influences its healthcare strategy regarding dependency, multimorbidity, and chronic conditions.

The case group included all patients who sustained a fall during hospitalisation within the study period, as recorded in their medical history and the standardised fall registry. The control group was selected from a list of patients admitted between 2020 and 2022, provided by the Admissions Department. Cases (patients with a recorded fall) were excluded, and a matching technique (1:1, exact match) was applied based on age, as it is recognised as one of the main intrinsic risk factors for falls [4]. Matching was also conducted by year of hospitalisation, to account for the potential influence of COVID-19 in 2020 and the patient isolation measures implemented during that period [21]. Each case was assigned one matched control, ensuring an exact match by age and year of hospitalisation.

To achieve a 1% precision in estimating the proportion of hospitalised patients who experience a fall during their stay, the sample size was calculated using an asymptotic normal confidence interval with finite population correction at a 95% bilateral confidence level. Assuming an expected incidence of 1.61 falls per 1000 patient-days [3]: a sample size of 391 individuals was deemed sufficient to ensure a 95% confidence level and a ±1 percentage point precision. A 5% replacement rate was anticipated to account for potential dropouts or exclusions.

### 2.3. Study Variables

Hospitalisation episode data were collected from electronic medical records, including sociodemographic and clinical variables such as age, sex, type of hospital unit (medical, surgical, or mixed), mean total length of stay, mean length of stay before the fall, and patient autonomy level. The latter was obtained from clinical documentation, where nurses perform and record a comprehensive standardised assessment of the patient based on Virginia Henderson’s Needs Model, considering whether the patient required assistance with feeding, dressing/personal grooming, bathing and hygiene, and/or mobility. Autonomy levels were classified as independent, requiring partial assistance, or requiring total assistance. This is a standardised practice that nurses perform in their daily practice, which is used to develop the care plan.

The following fall-related records were analysed:Fall risk assessment using the H.J. Downton Scale (1991) [22]. This validated scale evaluates previous falls, pharmacological treatment, sensory deficits, mental status, and ambulation, based on nurses’ clinical judgement. Scores range from 0 to 11, with a score ≥ 3 indicating a high risk of falling. The assessment is conducted within the first 24 h of admission and is reassessed in a standardised manner whenever there is a change in the patient’s condition or following a fall. For this study, the last recorded assessment before the fall was used for the case group, while for the control group, the last assessment prior to discharge was considered.The standardised fall incident report, which records the date, time, and circumstances surrounding the event.

Within the framework of the BPSO^®^ programme, quarterly audits are carried out on fall-related records to evaluate the completion of all fields in the standardised record. Those responsible for falls in each unit are then provided with feedback on areas for improvement. This helps to ensure that these records are accurate and complete.

DRGs and their resource consumption and cost estimators (relative weights) were provided by the Hospital’s Coding Unit. The average cost associated with each DRG was obtained from the official website of the Spanish Ministry of Health, which provides estimated relative weights for DRGs within the Spanish healthcare system. The 2022 estimates were used, as they were the most recent figures available at the time of the study [23].

### 2.4. Statistical Analysis

Quantitative variables are presented as means and standard deviations, while qualitative variables are expressed as frequency distributions. The association between qualitative variables was analysed using Pearson’s chi-squared test. When more than 20% of cells had expected values below 5, Fisher’s exact test or the Likelihood Ratio test (for variables with more than two categories) was applied.

Quantitative variables were compared using Student’s *t*-test for independent samples. Adjusted odds ratios (ORs) and their 95% confidence intervals (CIs) were calculated for a significance level of *p* < 0.05.

Data were analysed using IBM SPSS Statistics version 29.0 for Windows. A *p*-value < 0.05 was considered statistically significant.

## 3. Results

### 3.1. Description of the Sample and Study Groups

A total of 623 recorded falls were identified among 67,298 patients admitted to the 19 inpatient units between 2020 and 2022. The incidence of falls was 0.915%, with a fall rate of 1.43 per 1000 patient-days. Seven cases were excluded due to insufficient records, resulting in a final sample of 616 cases and 623 controls.

The mean age in both groups was 73.75 years (SD: 12.53 for cases; SD: 14.68 for controls), with no statistically significant difference (*p* = 0.996). The distribution of hospitalised patients across the study years was as follows:2020: Cases: n = 140 vs. Controls: n = 1442021: Cases: n = 227 vs. Controls: n = 2272022: Cases: n = 249 vs. Controls: n = 252

As shown in Table 1, the study population consisted predominantly of men in both groups (*p* = 0.033). The case group exhibited statistically significant differences compared to the control group (*p* < 0.001) in the following aspects: a higher frequency of previous falls; a greater level of dependency, with a higher proportion of patients requiring partial assistance; higher mean scores on the H.J. Downton Fall Risk Scale; and longer hospital stays.

A total of 203 different DRGs were identified across the 1239 hospitalisation episodes analysed.

Table 2 presents the most frequently identified DRGs, their distribution in both groups, and their associated relative weights.

### 3.2. Association Between DRGs and Risk of Falling

An association analysis was conducted to identify which DRGs were associated with a higher likelihood of falling. Only one DRG was found to be significantly associated: ‘Lower limb amputation except toes’. Hospitalisation episodes classified under this DRG had a fourfold increased risk of falling compared to all other DRGs (OR = 3.933; lower limit = 1.459; upper limit = 10.60). No other DRG was identified as a risk factor for falls.

### 3.3. Economic Analysis

To identify DRGs in which relative weights were higher than the standard in the case group compared to controls, odds ratios (ORs) and their 95% confidence intervals (CIs) were calculated. The analysis compared relative weights ≥ 2 vs. weights = 1 in both study groups for each DRG.

Additionally, since some DRGs had no hospitalisation episodes with a relative weight of 1 in either the case or control group, the analysis was repeated by grouping relative weights ≥ 3 vs. weights ≤ 2.

Five DRGs were identified in which having experienced a fall was significantly associated with an increase in relative weight and, consequently, with higher costs. See Table 3.

Based on the DRGs identified in Table 3, the estimated excess costs attributable to a fall were calculated by determining the difference between the relative weights ≥ 2 and the standard weight = 1 for each DRG. See Table 4.

From Table 3 and Table 4, it was observed that the probability of hospitalisation episodes in patients who had experienced a fall having a significantly higher relative weight was as follows:For ‘Urethral and transurethral procedures’ were associated with a 10.5 times higher probability of hospitalisation, with excess costs ranging from EUR 942.7983 to EUR 14,281.47. Three patients fell into the highest cost stratums (446-3).‘Heart valve procedures without AMI or complex diagnosis’ showed a 6.9 times higher probability, with an excess cost ranging from EUR 4461.87 to EUR 24,234.05. Twelve patients fell into the highest cost stratums (163-4 and 163-3).Arterial procedures on the lower limb’ had a 5-times higher probability, with an excess cost ranging from EUR 3578.81 to EUR 29,959.84. Twelve patients fell into the highest cost stratums (181-4 and 181-3).‘Heart failure’ had a 4.57 times higher probability, with an excess cost ranging from EUR 1030.88 to EUR 4167.02. Twenty patients fell into the highest cost stratums (194-4 and 194-3).‘Major pulmonary infections and inflammations’ had a 3.74 times higher probability, with an excess cost ranging from EUR 834.07 to EUR 6300.69. Thirty-three patients fell into the highest cost stratums (137-4 and 137-3).

The first three DRGs are surgical, while the last two are medical.

## 4. Discussion

The incidence of falls recorded in our hospital between 2020 and 2022, along with the fall rate per 1000 patient-days, was lower than that reported in similar studies conducted in Spanish hospitals [3]. This lower incidence rate is encouraging and aligns with the goals of the BPSO^®^ programme implemented at our institution, though a direct causal link cannot be established from this study. The recommendations from the BPG Preventing Falls and Reducing Injury from Falls, developed by the Registered Nurses’ Association of Ontario (RNAO^®^) have shown positive outcomes in studies conducted in some acute care hospitals in Spain [24].

When comparing the case and control groups, a higher frequency of previous falls within the last year was observed among cases, as well as higher fall risk scores recorded by nurses. These findings align with those reported in other studies [1]. Notably, the average length of stay for patients who had experienced a fall was significantly longer than for those who had not (21 days vs. 8 days). These two variables appear to be interdependent: the longer a patient remains hospitalised, the greater their fall risk, due to both the unfamiliarity of the environment and the progressive decline in health status. Conversely, experiencing a fall itself may contribute to prolonged hospitalisation [4]. In our study population, the length of stay before the fall (9 days) exceeded the overall average length of stay (8 days). Length of stay before falls could be considered an important risk factor to be taken into account in fall prevention models. This variable considers hospitalisation as a risk factor in itself that increases the risk of falls, frailty and cognitive weakness in older people, as some studies have shown [25]. These findings suggest that this variable could be predictive in the future, together with the patient’s comorbidity burden upon admission, as both factors could be related. As outlined preventive actions should be prioritised for patients who remain hospitalised beyond 8 days, and particularly those requiring partial assistance. We have not found any studies that support these findings, so further research is needed to corroborate this.

On the other hand, recent studies have identified falls as indicators of underliving patient complexity and may represent a marker, rather than a direct cause of increased complexity and cost, when occurring during hospitalisation [26].

One DRG was identified as having a fourfold increased risk of falls compared to the others: ‘Lower limb amputation except toes. A possible explanation is that these patients may not yet have fully adapted to their new level of dependency, transitioning from being independent to requiring partial assistance. As a result, they represent a high-risk group that should be targeted with specific care plans, including active participation in fall prevention strategies. The acquisition of new mobility skills through post-intervention education may require more time than is typically available during a hospital stay. This highlights an opportunity for improvement, as involving patients and their families in their own safety plan is a highly effective recommendation [27].

It is clear that the standard cost per hospitalised patient varies depending on the type of care required. Therefore, when analysing fall-related costs, it is essential not only to quantify direct expenses, but also to consider the entire hospitalisation episode, as classified by DRGs and their relative weights. Some studies have reported a higher incidence of falls in medical units compared to surgical units [28], while others highlight surgical procedures as a risk factor for falls during hospitalisation [29]. In our study, no significant differences were found in fall incidence between medical and surgical units. However, three predominantly surgical DRGs were identified in which falling was associated with increased hospital costs: ‘Urethral and transurethral procedures’; ‘Heart valve procedures without AMI or complex diagnosis’; and ‘Arterial procedures on the lower limb’. Additionally, two medical DRGs—‘Major pulmonary infections and inflammations’ and ‘Heart failure’—also showed increased costs in the case group, albeit to a lesser extent.

These findings suggest the need to closely monitor both medical and surgical services that contributed to these results. This will enable nurses to develop tailored strategies aimed at improving patient safety within these specific DRGs. In this regard, nurses are uniquely positioned to design individualised care plans, promote self-care, and implement targeted interventions to enhance patient safety [15]. On the other hand, as previously shown, DRG-based billing does not accurately capture the costs associated with nursing care. This study provides evidence that supports further investigation into a potential reform of the system, whereby the complexity of nursing care would be incorporated into cost accounting and billing during hospitalisation.

### 4.1. Limitations

There is a possibility of underreporting of falls. However, we believe this risk is minimised, as the BPSO^®^ programme requires a systematic evaluation of the Best BPG indicators. Additionally, at least one designated nurse in each inpatient unit is responsible for promoting adherence to the recommendations of the RNAO^®^ BPG Preventing Falls and Reducing Injury from Falls, including compliance with fall reporting and fostering an institutional culture of patient safety. Another limitation worth noting is the absence of functional status or comorbidity indices (e.g., Charlson Index or Elixhauser Comorbidity Index) that could help to more clearly specify some of the differences in costs. Finally, although assessing the ‘level of dependency’ is common practice in nursing, its subjectivity could introduce variability.

### 4.2. Recommendations for Further Research

The researchers try to foster further research with large populations including other hospitals in the Castile and Leon Community with similar characteristics, in order to obtain more conclusive results. In next studies we will include factors like length of stay as a risk factor for falls and increased costs, and baseline comorbidity or severity of illness in the case–control matching.

### 4.3. Implications for Policy and Practice

Analysing the costs of falls based on DRGs among hospitalised patients will enable nurses and managers to design effective prevention strategies to improve patient safety. These findings support nurses responsible for implementing the BPG Preventing Falls and Reducing Injury from Falls, developed by the Registered Nurses’ Association of Ontario, in designing specific nursing management strategies for the identified DRGs. Nurse-led management of fall prevention strategies could contribute to a more sustainable healthcare system.

From a practical standpoint, it seems clear that higher-weight DRGs, which have shown a greater probability of falls, are frequently associated with episodes of hospitalisation involving elevated care complexity, prolonged lengths of stay, invasive interventions, or the management of clinical complications. These conditions intrinsically generate greater demand for nursing resources. Therefore, the billing model should take into account the complexity of nursing activities, as this could lead to improved nursing management strategies and better patient outcomes.

## 5. Conclusions

Patients classified under the DRG ‘Lower limb amputation except toes’ had a fourfold increased risk of falling compared to other DRGs.

Five DRGs were identified in which the probability of hospitalisation episodes involving a fall was associated with higher economic costs (relative weight) compared to episodes without a fall: three surgical DRGs (‘Urethral and transurethral procedures’; ‘Heart valve procedures without AMI or complex diagnosis’ and ‘Arterial procedures on the lower limb’) and two medical DRGs (‘Heart failure’ and ‘Major pulmonary infections and inflammations’).

Identifying DRGs in which falls were associated with increased hospitalisation costs provides a comprehensive approach to cost assessment. This includes not only direct fall-related expenses but also the broader impact in terms of resource consumption and clinical characteristics of hospitalised patients.

## Figures and Tables

**Table 1 nursrep-15-00323-t001:** Descriptive data of the case and control groups.

		Cases (n = 616)		Controls (623)		*p*-Value
		n	Relative %	n	Relative %	
Sex	Male	416	67.53	381	61.15	0.033
	Female	200	32.46	242	38,84	
Previous falls	Yes	41	6.65	25	4.01	<0.001
	No	575	93.34	598	95.98	
Level of dependency	Independent	37	6.00	73	11.71	<0.001
	Partial assistance	77	12.5	45	7.22	
	Total assistance	26	4.22	25	4.05	
Hospital unit	Medical	271	54.20	229	45.80	0.235
	Surgical	282	46.80	320	53.10	
	Mixed	63	46.60	72	53.30	
		**Mean**	**SD ***	**Mean**	**SD ***	
H.J. Downton Fall Risk Score		3.26	2.05	2.69	1.64	<0.001
Mean length of stay (days)		21.26	21.49	8.05	10.23	<0.001
Length of stay before fall (days)		9.95	15.14	-	-	

SD * = standard deviation.

**Table 2 nursrep-15-00323-t002:** Descriptive data of the most frequent diagnosis-related groups (DRGs) and their relative weight in the two study groups.

Diagnosis-Related Group (DRG) (Total n)	DRG Code-Weight				
		Control Group		Case Group	
		n	%	n	%
Major pulmonary infections and inflammations (n = 81)	137-2	17	40.5%	6	15.4%
	137-3	21	50.0%	21	53.8%
	137-4	4	9.5%	12	30.8%
	137	42	100%	39	100%
Heart failure (n = 55)	194-1	2	6.7%	0	0.0%
	194-2	14	46.7%	5	20.0%
	194-3	11	36.7%	11	44.0%
	194-4	3	10.0%	9	36.0%
	194	30	100%	25	100%
Arterial procedures on the lower limb (n = 43)	181-1	10	55.6%	5	20.0%
	181-2	6	33.3%	8	32.0%
	181-3	2	11.1%	11	44.0%
	181-4	0	0.0%	1	4.0%
	181	18	100%	25	100%
Chronic obstructive pulmonary disease (n = 35)	140-2	2	11.1%	1	5.9%
	140-3	10	55.6%	11	64.7%
	140-4	6	33.3%	5	29.4%
	140	18	100%	17	100%
Percutaneous coronary interventions without AMI * (n = 35)	175-1	4	22.2%	1	5.3%
	175-2	4	22.2%	3	15.8%
	175-3	3	16.7%	6	31.6%
	175-4	5	27.8%	9	47.4%
	175	16	100%	19	100%
Other pneumonia (n = 35)	139-1	1	7.7%	4	18.2%
	139-2	3	23.1%	6	27.3%
	139-3	5	38.5%	7	31.8%
	139-4	4	30.8%	5	22.7%
	139	13	100%	22	100%
Sepsis and disseminated infections (n = 28)	720-1	1	10.0%	0	0.0%
	720-2	2	20.0%	5	27.8%
	720-3	7	70.0%	6	33.3%
	720-4	0	0.0%	7	38.9%
	720	10	100%	18	100%
Heart valve procedures without AMI * or complex diagnosis (n = 27)	163-1	2	18.2%	0	0.0%
	163-2	6	54.5%	4	25.0%
	163-3	3	27.3%	8	50.0%
	163-4	0	0.0%	4	25.0%
	163	11	100%	16	100%
Lower limb amputation except toes (n = 24)	305-1	1	20.0%	2	10.5%
	305-2	4	80.0%	12	63.2%
	305-3	0	0.0%	4	21.1%
	305-4	0	0.0%	1	5.3%
	305	5	100%	19	100%
Kidney and urinary tract infections (n = 24)	463-1	3	18.8%	1	12.5%
	463-2	3	18.8%	2	25.0%
	463-3	8	50.0%	4	50.0%
	463-4	2	12.5%	1	12.5%
	463	16	100%	8	100%
Percutaneous coronary interventions with AMI * (n = 23)	174-1	1	12.5%	1	6.7%
	174-2	4	50.0%	6	40.0%
	174-3	3	37.5%	2	13.3%
	174-4	0	0.0%	6	40.0%
	174	8	100%	15	100%
ACVA ** and precerebral occlusions with infarction (n = 21)	045-1	1	8.3%	1	11.1%
	045-2	8	66.7%	2	22.2%
	045-3	3	25.0%	4	44.4%
	045-4	0	0.0%	2	22.2%
Peripheral vascular disorders and others (n = 20)	197-1	0	0.0%	1	12.5%
	197-2	6	50.0%	3	37.5%
	197-3	5	41.7%	4	50.0%
	197-4	1	8.3%	0	0.0%
	197	12	100%	8	100%
Pancreatic disorders except malignant neoplasm (n = 19)	282-1	3	30.0%	1	11.1%
	282-2	5	50.0%	3	33.3%
	282-3	2	20.0%	4	44.4%
	282-4	0	0.0%	1	11.1%
	282	10	100%	9	100%
Biliary tract and gallbladder disorders (n = 19)	284-1	3	37.5%	2	18.2%
	284-2	3	37.5%	3	27.3%
	284-3	2	25.0%	6	54.5%
	284	8	100%	12	100%
Pulmonary embolism (n = 18)	134-1	4	40.0%	0	0.0%
	134-2	2	20.0%	3	375%
	134-3	4	40.0%	3	37.5%
	134-4	0	0.0%	2	25.0%
	134	10	100%	8	100%
Respiratory neoplasms (n = 18)	136-1	1	10.0%	0	0.0%
	136-2	3	30.0%	1	12.5%
	136-3	6	60.0%	6	75.0%
	136-4	0	0.0%	1	12.5%
	136	10	100%	8	100%
Permanent cardiac pacemaker implantation without AMI *, heart failure, or shock (n = 17)	171-1	2	16.7%	0	0.0%
	171-2	8	66.7%	4	80.0%
	171-3	1	8.3%	1	20.0%
	171-4	1	8.3%	0	0.0%
	171	12	100%	5	100%
Urethral and transurethral procedures (n = 17)	446-1	7	77.8%	2	25.0%
	446-2	1	11.1%	3	37.5%
	446-3	1	11.1%	3	37.5%
	446	9	100%	8	100%

AMI * = acute myocardial infarction; ACVA ** = acute cerebrovascular accident.

**Table 3 nursrep-15-00323-t003:** Association between number of falls (cases) and DRGs with relative weight ≥2 vs. =1 and DRGs with relative weight ≥ 3 vs. ≤2 (statistically significant results).

		Group			95% CI	
DRG (DRG Code)	Weight	Case	Control	OR	Lower Limit	Upper Limit
Urethral and transurethral procedures (code 446)	≥2	6	2	10.50	1.11	98.92
	=1	2	7			
Total		8	9			
Arterial procedures on the lower limb (code 181)	≥2	20	8	5.00	1.30	19.30
	=1	5	10			
Total		25	18			
Heart valve procedures without AMI * or complex diagnosis (code 163)	≥3	13	3	6.933	1.291	37.225
	≤2	5	8			
Total		18	11			
Heart failure (code 194)	≥3	20	14	4.571	1.357	15.399
	≤2	5	16			
Total		25	30			
Major pulmonary infections and inflammations (code 137)	≥3	33	25	3.74	1.288	10.860
	≤2	6	17			
Total		39	42			

AMI * = acute myocardial infarction.

**Table 4 nursrep-15-00323-t004:** Attributable excess cost of a fall in DRGs with statistically significant higher relative weights (cost difference compared to the standard cost or weight = 1).

DRG	DRG Code-Weight	Standard Cost (€)	Excess Cost Attributable to a Fall (€)
Urethral and transurethral procedures	446-1	2459.99	
	446-2	3402.82	942.83
	446-3	7064.27	4604.28
	446-4	16,741.42	14,281.47
Arterial procedures on the lower limb	181-1	10,610.14	
	181-2	15,072.02	4461.87
	181-3	23,053.96	12,443.82
	181-4	34,844.19	24,234.05
Heart valve procedures without AMI or complex diagnosis	163-1	18,905.62	
	163-2	22,484.43	3578.81
	163-3	31,765.50	12,859.87
	164-4	48,865.46	29,959.84
Heart failure	194-1	2621.17	
	194-2	3652.05	1030.88
	194-3	4487.40	1866.23
	194-4	6788.19	4167.02
Major pulmonary infections and inflammations	137-1	3324.10	
	137-2	4158.17	834.07
	137-3	5078.99	1754.89
	137-4	9624.79	6300.69

## Data Availability

Restrictions apply to the availability of this data. The data was obtained from patient medical records and is available upon request to the lead author with the authorisation of the Valladolid Health Area Ethics Committee and the Castile and León Regional Health Authority.

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
