# Peer review of "Costs Attributable to Falls Based on Diagnosis-Related Groups (DRGs) Analysis of Hospitalised Patients: A Case–Control Study"

_nursrep, 2025, doi:10.3390/nursrep15090323_

Round 1

Reviewer 1 Report (Previous Reviewer 1)

Comments and Suggestions for Authors

The manuscript has improved substantially compared to an earlier draft. In my opinion, these are the remaining issues:

1) In the opening line of the abstract, the word “Bacground/objetives” appears. Is this a typographical error that should be corrected to “Background/Objectives”? In the main text there are terms such as “indentificated” instead of “identified” and “Recomendaciones” instead of “Recommendations.” The manuscript would benefit from a careful language revision to improve the overall quality of English and to correct the typographical errors that are still present. Have the authors considered a final English language polish to ensure consistency throughout?
2) Some tables and captions show minor inconsistencies (e.g., “total” vs. “Total,” spacing, capitalization). Would the authors consider harmonizing formatting to comply with journal style?
3) The conclusions section currently restates several detailed results. Would the authors consider making this section more concise and focusing mainly on the implications for practice and policy?
4) The manuscript notes that DRG-based funding does not directly capture nursing care costs. Could the authors strengthen this argument to better highlight how their findings provide evidence for system-level change?
5) The manuscript contains both an “Ethical Considerations” paragraph and an “Institutional Review Board Statement,” which partly repeat the same information. Would the authors consider merging these into a single, streamlined statement to avoid redundancy?
6) The waiver of informed consent appears justified given the retrospective design, the absence of direct patient contact, and the anonymization of data. Could the authors briefly clarify in the text that the Ethics Committee explicitly granted this waiver, rather than implying it was assumed?

Comments on the Quality of English Language

The manuscript is generally understandable; however, the quality of English requires improvement. Several typographical and grammatical errors are present. A thorough language edit would help enhance clarity, consistency, and readability. I recommend that the authors consider professional English editing prior to final submission.

Author Response

Response to Reviewer 1

We would like to reiterate our gratitude for your invaluable guidance in improving our work.

1) In the opening line of the abstract, the word “Bacground/objetives” appears. Is this a typographical error that should be corrected to “Background/Objectives”? In the main text there are terms such as “indentificated” instead of “identified” and “Recomendaciones” instead of “Recommendations.” The manuscript would benefit from a careful language revision to improve the overall quality of English and to correct the typographical errors that are still present. Have the authors considered a final English language polish to ensure consistency throughout?

We thank your comments regarding the typographical errors found in the text. We believe these were caused by Word’s autocorrect during transcription, and we apologize for not having noticed them.

We have reviewed the text and hope there are no further errors. If there are, please let us know. We appreciate your patience.

We have also attached the Language Editing Certificate.

2) Some tables and captions show minor inconsistencies (e.g., “total” vs. “Total,” spacing, capitalization). Would the authors consider harmonizing formatting to comply with journal style?

Thank you for your advice. We have reviewed the tables, spacing and capitalization

3) The conclusions section currently restates several detailed results. Would the authors consider making this section more concise and focusing mainly on the implications for practice and policy?

The conclusions have been summarized according to your suggestions

4) The manuscript notes that DRG-based funding does not directly capture nursing care costs. Could the authors strengthen this argument to better highlight how their findings provide evidence for system-level change?

Thank you for your suggestion. We have added a comment in lines 323-327.

5) The manuscript contains both an “Ethical Considerations” paragraph and an “Institutional Review Board Statement,” which partly repeat the same information. Would the authors consider merging these into a single, streamlined statement to avoid redundancy?

Thank you for your recommendation. We have merged the two sections into one. Lines 383-398

6) The waiver of informed consent appears justified given the retrospective design, the absence of direct patient contact, and the anonymization of data. Could the authors briefly clarify in the text that the Ethics Committee explicitly granted this waiver, rather than implying it was assumed?

This information has been clarified in lines 384 and 387

Thank you very much for your kind attention

Reviewer 2 Report (Previous Reviewer 2)

Comments and Suggestions for Authors

Satisfied with changes

Author Response

Response to Reviewer 2

We would like to reiterate our gratitude for your invaluable guidance in improving our work.

Thank you for your kind attention.

Kind regards

This manuscript is a resubmission of an earlier submission. The following is a list of the peer review reports and author responses from that submission.

Round 1

Reviewer 1 Report

Comments and Suggestions for Authors

Dear Authors, congratulations for your scientific work. A few areas require clarification or slight refinement. Below are specific comments:

1) MDPI style needs to be incorporated in the references section (https://www.mdpi.com/authors/references). Please, modify it and ensure most sources (>90%) are from the past five years to reflect the topic’s timeliness.

2) The abstract is well-structured and informative. However, the section “Background/objectives” is inconsistently formatted and should be corrected. Please review this text carefully, as there are numerous typographical issues. You may consider breaking the aims into a concise sentence and clarifying the economic indicator (relative weight) earlier. For example: "This study aimed to estimate excess hospitalisation costs attributable to inpatient falls, using Diagnosis-Related Group (DRG) relative weights as a proxy for resource consumption."

3) The rationale for using DRGs as a proxy for costs is appropriate. However, additional explanation on how DRG weights are operationalised in the Spanish context (e.g., update frequency, role in hospital funding) would benefit international readers unfamiliar with the system. Current nursing research has highlighted the use of DRG weight as a proxy for costs and its direct relationship with nursing care complexity. Some parallels are needed and should be discussed.

4) Matching controls on age and year is reasonable, but readers would benefit from a clearer explanation of whether matching was performed individually (1:1 exact match) or frequency-based. This need to be clarified.

5) You mention using “relative weight ≥2 vs. =1” and “≥3 vs. ≤2” as thresholds. How were these cutoffs determined? Were they based on distributional percentiles, clinical reasoning, or precedent in the literature? Please clarify it.

6) Autonomy levels are based on clinical judgement following Virginia Henderson’s model. This approach is valid but inherently subjective. Please clarify whether inter-rater agreement was assessed or whether this was retrieved from nursing documentation.

7) The statistical presentation is complete and clear. Tables are extensive but informative. Good!

8) You might consider collapsing some less relevant DRGs into an appendix to streamline the main results section.

9) The relative weight odds ratios are compelling. However, the interpretation could be enhanced by explaining how many patients (in absolute terms) fell into the highest-cost strata (e.g., DRG 446-4).

10) The discussion interprets the data coherently. One sentence could be added to explicitly acknowledge that falls may represent a marker, rather than a direct cause, of increased complexity and cost, even when occurring prior to discharge. This interpretation is consistent with recent studies that have examined falls as indicators of underlying patient complexity.

11) The association between “length of stay before fall” and overall LOS is a strength and deserves slightly more discussion. Could this variable be predictive in a future prospective risk model? Please, provide some insights about it.

12) The implication that prevention efforts should be targeted after 8 days of hospitalisation is a valuable insight—have similar thresholds been suggested in existing guidelines or studies? This point needs to be clarified.

13) You rightly mention the potential underreporting of falls. Another limitation worth noting is the absence of functional status or comorbidity indices (e.g., Charlson or Elixhauser) which could help explain confounding differences in cost. This is an important point to be considered for the limitations section.

14) Kindly indicate the date on which the ethics committee granted approval, to be included in the Institutional Review Board Statement. Furthermore, please include a separate paragraph in the manuscript addressing the ethical aspects of the study, clearly stating the ethics committee approval and the procedures for obtaining informed consent. The retrospective nature of the study does not in itself justify the omission of informed consent, which should still be addressed and appropriately justified.

Some additional specific questions:

1) How did you decide on the DRG thresholds (≥2, ≥3)? Would alternative stratifications (e.g., quartiles or cost quintiles) yield different patterns?

2) Could you provide additional information on how nurse-reported fall incidents are validated or monitored for completeness?

3) Have you considered including an analysis of length of stay as a mediating variable between fall and increased cost?

4) Given that “Lower limb amputation” is a high-risk DRG, could the fall risk be mitigated through standardised early mobility protocols? Has this been explored in your setting?

Author Response

Point-by-point response to reviewer 1

Manuscript ID: nursrep-3789678

Title: Costs Attributable to Falls Based on Diagnosis-Related Groups (DRGs) Analysis of Hospitalised Patients: A Case-Control Study

Authors: Mercedes Fernández-Castro, Noel Rivas-González *, Belén Martín-Gil, Pedro Luis Muñoz-Rubio, Rocío Lozano-Pérez, Pilar Rodríguez-Soberado, M. Fe Muñoz Moreno

Dear editor of Nursig Reports

We want to express our sincere appreciation and thanks for your swift response in the evaluation of our work. We are sure that all of reviewers’ comments improve the content of our work without any doubt. Accordingly, we have made the changes that reviewers have suggested in their evaluation. This changes are written in red in the text.

The responses and explanations regarding reviewers’ issues are developed below:

Reviewer 1 comments / Authors response to Reviewer 1:

Thank you very much for your suggestions and comments. We are grateful to you for detailing the points of our work that need improvement. We have considered the issues you suggest, carefully.

The changes according your recommendations have been written in red in the text:

1) MDPI style needs to be incorporated in the references section (https://www.mdpi.com/authors/references). Please, modify it and ensure most sources (>90%) are from the past five years to reflect the topic’s timeliness.

Thanks for your comment. References have been changed according to MDPI style. In addition, references number 2,7,15 and 22 have been replaced by more recent ones and some comments added to the text have been supported by new references written in red in the text (2,7,13,15,18,24 and 25)

2) The abstract is well-structured and informative. However, the section “Background/objectives” is inconsistently formatted and should be corrected. Please review this text carefully, as there are numerous typographical issues. You may consider breaking the aims into a concise sentence and clarifying the economic indicator (relative weight) earlier. For example: "This study aimed to estimate excess hospitalisation costs attributable to inpatient falls, using Diagnosis-Related Group (DRG) relative weights as a proxy for resource consumption."

Thank you very much for your comment. The objective has been changed in abstract section as you suggested, as it is now clearer (Lines 23-25).

3) The rationale for using DRGs as a proxy for costs is appropriate. However, additional explanation on how DRG weights are operationalised in the Spanish context (e.g., update frequency, role in hospital funding) would benefit international readers unfamiliar with the system. Current nursing research has highlighted the use of DRG weight as a proxy for costs and its direct relationship with nursing care complexity. Some parallels are needed and should be discussed.

Thank you for your suggest, we have included additional explanation about how DRG weingt are operationalized in the Spanish context in the paragraph (Lines 72-87)

A reference supporting this information has also been attached, along with the link to the Spanish Ministry of Health

Regarding the relationship of this billing system with the complexity of care provided by nurses, it is important to highlight that unfortunately, no recognized healthcare billing system includes the complexity of nursing care. We appreciate your comment on this point and have added it in the section 'Implications for policy and practice” (Lines 349-355)

4) Matching controls on age and year is reasonable, but readers would benefit from a clearer explanation of whether matching was performed individually (1:1 exact match) or frequency-based. This need to be clarified.

Thank you for requesting clarification, matching was carried out individually (1:1, exact match), between each case and its corresponding control, ensuring an exact match by age and year of hospitalisation. (Line 135 and 139)

5) You mention using “relative weight ≥2 vs. =1” and “≥3 vs. ≤2” as thresholds. How were these cutoffs determined? Were they based on distributional percentiles, clinical reasoning, or precedent in the literature? Please clarify it.

This point is exposed between lines 92-94 and 230-236. If you feel that it is not sufficiently clear, please let us know.

The DRG system assigns a relative weight based on stratified cost levels according to the severity of the hospitalisation episode: minor (weight=1), moderate (Weight=2), major (weight=3), and extreme (Weight=4).

All episodes with weights greater than the standard (weight=1) for the identified GRDs are compared. “Relative weight ≥2 vs. = relative weight 1”.

when there are no patients with Relative weight = 1 in a DRG we compared ““relative weight ≥3 vs. Relative weight ≤2

6) Autonomy levels are based on clinical judgement following Virginia Henderson’s model. This approach is valid but inherently subjective. Please clarify whether inter-rater agreement was assessed or whether this was retrieved from nursing documentation.

Thank you for your comment. We have clarified this point (Lines 153-159) and also has been included in the limitations as a potential bias (Lines330-334). Although nurses use their clinical judgement to assess the patient's level of autonomy, they perform and record a comprehensive standardized assessment of the patient based on Virginia Henderson's needs. No inter-rater reliability assessment has been performed because this is a standardised practice that nurses carry out in their daily practice, on the basis of which the care plan is developed, while this is a common nursing practice, its subjectivity can introduce variability, so we consider it like a limitation.

7) The statistical presentation is complete and clear. Tables are extensive but informative. Good!

Thank you for your comment

8) You might consider collapsing some less relevant DRGs into an appendix to streamline the main results section.

The authors kindly request that you reconsider this suggestion. Table 2 is very detailed and serves as a descriptive foundation, showing the distribution of patients across of the most Frequent Diagnosis-Related Groups (DRGs), it is important for us to strengthen the paper's transparency.

9) The relative weight odds ratios are compelling. However, the interpretation could be enhanced by explaining how many patients (in absolute terms) fell into the highest-cost strata (e.g., DRG 446-4).

Thank you for your comment. Table 2 presents the number of patients in each DRG category. In response to your suggestion, we specifically extracted data for DRGs with a statistically significant higher likelihood of falls which fell into the highest-cost strata. (Lines 253-266)

10) The discussion interprets the data coherently. One sentence could be added to explicitly acknowledge that falls may represent a marker, rather than a direct cause, of increased complexity and cost, even when occurring prior to discharge. This interpretation is consistent with recent studies that have examined falls as indicators of underlying patient complexity.

Thank you very much for that observation, we have included a paragraph considering falls as indicators of underlying patient complexity and a new reference to support this (Hagino et al, 2025). (Lines 295-297)

11) The association between “length of stay before fall” and overall LOS is a strength and deserves slightly more discussion. Could this variable be predictive in a future prospective risk model? Please, provide some insights about it.

Thank you for your comment, we appreciate your contribution to this reflection, we consider it important to emphasize the idea that being hospitalized is a risk factor in itself, as supported by the study by Seń M. et al. in 2022. We provide an explanation on this point in lines 286-291.

12) The implication that prevention efforts should be targeted after 8 days of hospitalisation is a valuable insight—have similar thresholds been suggested in existing guidelines or studies? This point needs to be clarified.

This comment responds to the fact that, in our study, the hospital length of stay before fall was longer than the average stay (8 days). We have not found any studies that support these findings, so further research is needed to corroborate this. We mentioned this in lines 293-294.

13) You rightly mention the potential underreporting of falls. Another limitation worth noting is the absence of functional status or comorbidity indices (e.g., Charlson or Elixhauser) which could help explain confounding differences in cost. This is an important point to be considered for the limitations section.

Thank you very much for your appreciation, in light of our results, we have also reflected on this question and completely agree with your comments. We are currently designing a new study in which factors such as burden of comorbidities, length of hospital stay, and the medical department in which patients were admitted, among others, will be taken into account when matching the case and control groups. In the meantime, we appreciate your understanding and would like to point out that we have included this aspect in the section on limitations and in the section on future studies. (Lines 330-334 and 349-355)

14) Kindly indicate the date on which the ethics committee granted approval, to be included in the Institutional Review Board Statement. Furthermore, please include a separate paragraph in the manuscript addressing the ethical aspects of the study, clearly stating the ethics committee approval and the procedures for obtaining informed consent. The retrospective nature of the study does not in itself justify the omission of informed consent, which should still be addressed and appropriately justified.

Thank you for your comment, we have included a new section “Ethical considerations” (Line 378-392), and the date on which the ethics committee granted approval  in the Institutional Review Board Statement (line 405).

Some additional specific questions:

1) How did you decide on the DRG thresholds (≥2, ≥3)? Would alternative stratifications (e.g., quartiles or cost quintiles) yield different patterns?

We consider that this point has been clarified in question 5. If you feel that it is not sufficiently clear, please let us know.

2) Could you provide additional information on how nurse-reported fall incidents are validated or monitored for completeness?

Thank you for your comment, we have included additional information on how nurse-reported fall incidents are validated or monitored for completeness, after variables section (Lines 171-174).

3) Have you considered including an analysis of length of stay as a mediating variable between fall and increased cost?

We consider this observation very interesting. Based on the idea that hospitalization itself constitutes a risk, we are designing a new study taking this key variable into account. We have included this in “in Further research” section (Line 338-340)

4) Given that “Lower limb amputation” is a high-risk DRG, could the fall risk be mitigated through standardised early mobility protocols? Has this been explored in your setting?

In our setting, affected patients and their families are given guidelines on how to obtain active participation in fall prevention strategies and the acquisition of new mobility skills through post-intervention education. Despite the acquisition of new mobility skills through post-intervention education may require more time than is typically available during a hospital stay, specific recommendations are given upon discharge (lines 303-306)

Reviewer 2 Report

Comments and Suggestions for Authors

  1. Main research question

The primary research question is explicitly and clearly articulated in the abstract. The authors stated their aim was "to analyse Diagnosis-Related Groups and their associated relative weight as an estimator of resource consumption and costs in hospitalised patients who sustained a fall, compared with a control group of non-fallers, in order to identify the excess costs attributable to a fall" (Lines 24-28). This objective was consistently pursued throughout the manuscript, guiding the methodology, results, and discussion.

  1. Originality and relevance

While the high cost of falls is a well-established fact in healthcare literature, this paper's originality lies in its specific approach. The key contributions to the field are:

  • Use of DRG relative weights as a cost proxy: Many studies calculate the direct costs associated with a fall (e.g., cost of surgery for a fracture, extra days in the hospital). This study’s method of analysing the shift in DRG relative weight (a measure of case complexity and resource use) between fallers and non-fallers within the same primary diagnosis is more sophisticated. It captured the overall increase in the complexity and cost of the entire hospitalisation episode that is precipitated by a fall, not just the direct expense of the fall-related injury.
  • Identification of high-risk DRGs: The study moved beyond generic risk factors (such as age) and identified a specific clinical DRG—'Lower limb amputation except toes'—as conferring a fourfold increased risk of falling (Lines 194-196). This is a powerful, actionable finding for clinical practice, suggesting that this patient cohort requires uniquely targeted fall prevention protocols.
  • Specific economic impact within DRGs: The identification of five specific DRGs (three surgical, two medical) where falls were associated with significantly higher costs is the paper's core strength (Lines 205-207). This provides granular data that hospital administrators and nurse managers can use to focus quality improvement and resource management efforts. For example, knowing that a fall in a patient undergoing a 'Heart valve procedure' (Line 224) has a disproportionately large financial impact can justify investment in specialised prevention strategies for that unit.
  1. Methodological improvements

The study design is generally sound, but several significant improvements should be considered to enhance the validity and robustness of the findings.

  • Outdated financial data: In lines 147-149, the authors stated, "The 2020 estimates were used, as they were the most recent figures available at the time of the study." This is a major limitation for a paper submitted for publication in 2025 that analysed data up to 2022. Global healthcare costs and inflation have risen substantially since 2020, particularly in the post-pandemic era. Using 2020 cost figures to calculate the "excess cost" in Table 4 for falls occurring in 2021 and 2022 significantly underestimates the true financial impact in contemporary terms.
    • Justification: The Spanish Ministry of Health does provide updated DRG weight and cost data periodically. A web search reveals that datasets are often released with a lag, but data from 2022 should be available or imminently available by the time of publication. A 2024 study by Gonzalez et al. on hospital costs, for example, would likely be expected to use data from at least 2022 if available.
    • Recommendation: The authors must perform and report on a thorough search for updated cost data from the Spanish Ministry of Health. If more recent data (e.g., for 2022) is available, the analysis in Table 4 should be redone. If it is absolutely unavailable, this must be stated as a primary limitation in the discussion. The authors should then re-frame the costs in Table 4 as "Estimated Excess Cost (in 2020 Euros)" and explicitly caution readers that these figures do not account for subsequent healthcare inflation. The odds ratios (ORs) for increased relative weights in Table 3 would remain valid, but the absolute monetary values are questionable.
  • Control group matching and confounding variables: The cases and controls were matched by age and year of hospitalisation (Lines 113-114). While essential, this is insufficient to control for critical confounding variables. The primary driver of hospital length of stay and resource consumption is the patient's baseline severity of illness and burden of comorbidities. A relatively healthy 75-year-old admitted for a straightforward procedure is fundamentally different from a 75-year-old with multiple chronic conditions admitted for the same procedure.
    • Justification: It is a standard in high-quality observational research on hospital outcomes to adjust for baseline health status using validated tools such as the Charlson Comorbidity Index or by controlling for the initial admission DRG severity. The fact that the case group had a longer length of stay before the fall (9.95 days vs. 8.05 days total for controls) (Line 180) strongly suggests the groups were not comparable at baseline; the case group was likely sicker on admission.
    • Recommendation: The authors should acknowledge this as a significant limitation. Ideally, given the retrospective nature of the study, they could extract comorbidity data from the records and perform a multivariate logistic regression analysis, adjusting the odds ratios for a comorbidity index. This would provide a much more accurate estimate of the true effect of a fall, disentangled from pre-existing health status. At a minimum, a detailed discussion of this potential confounding is required.
  • Subjectivity of "level of dependency" assessment: The patient autonomy level was determined "using clinical judgement by the nurse" based on Virginia Henderson's Needs Model (Lines 128-129). While this is a common nursing practice, its subjectivity can introduce variability.
    • Recommendation: The authors should clarify if any measures were taken to ensure inter-rater reliability in these assessments. If not, this minor source of potential bias should be noted in the limitations section.
  1. Consistency of conclusions

The stated conclusions (Lines 302-318) are consistent with the evidence and arguments presented in the results section. The authors did not overstate their findings and directly addressed the core questions posed at the beginning of the study.

  • The conclusion that "Patients classified under the DRG ‘Lower limb amputation except toes’ had a four-fold increased risk of falling" was directly supported by the odds ratio reported in the analysis of DRGs and fall risk (Lines 194-196).
  • The conclusion that five specific DRGs were associated with higher economic costs after a fall was directly supported by the odds ratios presented in Table 3 and the subsequent cost calculations in Table 4.
  • The manuscript successfully answered its main question by identifying specific DRGs where falls led to excess costs, thereby providing a clear path from analysis to actionable clinical and administrative strategies.
  1. Comments on tables and data quality

The tables are information-rich and effectively present the study's data.

  • Table 1: Provides a clear and comprehensive demographic and clinical comparison of the case and control groups.
  • Table 2: This table is very detailed and serves as an excellent descriptive foundation, showing the distribution of patients across numerous DRGs. The level of detail strengthens the paper's transparency.
  • Table 3: This is the pivotal table for the economic analysis. The use of odds ratios to compare relative weight categories (e.g., ≥2 vs. =1) is an appropriate and effective statistical approach. The results are presented clearly with confidence intervals.
  • Table 4: As discussed in Point 3, the format of this table is clear, but the quality of the underlying data is compromised by the use of 2020 cost figures. The header, "Excess Cost Attributable to a Fall (€)," implies a certainty and timeliness that the data does not possess.

The overall quality of the raw data collection appears high, with a large sample size and detailed recording of clinical variables. The main weakness is not in the collection but in the external financial data used for the final calculations.

  1. Caveats and weaknesses

Several weaknesses beyond those already mentioned should be addressed to improve the manuscript.

  • Underdeveloped limitations section: The current limitations section (Lines 283-289) is far too brief. It mentioned only the "possibility of underreporting of falls." While the authors rightly argued this was minimised by their hospital's BPSO® program, a robust discussion of limitations is a cornerstone of good research. This section must be expanded to include:
    1. The use of outdated 2020 financial data (the most critical limitation).
    2. The single-center nature of the study, which limits generalisability to other hospitals with different patient populations or staffing models.
    3. The crucial limitation of not controlling for baseline comorbidity or severity of illness in the case-control matching.
    4. The potential subjectivity of the nursing assessment for dependency level.
  • Strength of claim regarding BPG program effectiveness: In lines 237-239, the authors suggested their lower fall rate "may reflect the effectiveness of the recommendations from the BPG." While plausible and encouraging, this observational study was not designed to prove the effectiveness of the BPG program. This wording should be softened to avoid implying causality. It is an association observed in a single center with an active program. A more cautious phrasing such as, "This lower incidence rate is encouraging and aligns with the goals of the BPG program implemented at our institution, though a direct causal link cannot be established from this study," would be more appropriate.
  • Insightful pre-fall length of stay: The finding that the average length of stay before a fall (9.95 days) already exceeded the total average stay for non-fallers (8.05 days) was a profoundly important point (Line 180) and was correctly highlighted in the discussion (Lines 253-254). This strongly supports the hypothesis that the case group was inherently more complex or ill upon admission and reinforces the need to control for this confounding variable. The authors should give this finding even more prominence as it fundamentally frames the entire comparison.

Author Response

Point-by-point response to reviewer 2

Manuscript ID: nursrep-3789678

Title: Costs Attributable to Falls Based on Diagnosis-Related Groups (DRGs) Analysis of Hospitalised Patients: A Case-Control Study

Authors: Mercedes Fernández-Castro, Noel Rivas-González *, Belén Martín-Gil, Pedro Luis Muñoz-Rubio, Rocío Lozano-Pérez, Pilar Rodríguez-Soberado, M. Fe Muñoz Moreno

Dear editor of Nursig Reports

We want to express our sincere appreciation and thanks for your swift response in the evaluation of our work. We are sure that all of reviewers’ comments improve the content of our work without any doubt. Accordingly, we have made the changes that reviewers have suggested in their evaluation. This changes are written in red in the text.

The responses and explanations regarding reviewers’ issues are developed below:

 Reviewer 2 comments / Authors response to Reviewer 2:

Thank you very much for your kinds comments and your suggestions. We are really grateful to you for detailing the points of our work that need improvement. We have considered all the issues you suggest, carefully.

We have corrected each of your recommendations and written them in red in the text.

  1. Methodological improvements

The study design is generally sound, but several significant improvements should be considered to enhance the validity and robustness of the findings.

Outdated financial data

Thank you for your comment, we appreciate your contribution to this reflection.

We have explored the Ministry of Health's website, where we found that the DRG costs for 2022 have been updated. Table 4 has been updated to reflect these changes, and the text has been corrected throughout.

Control group matching and confounding variables

Thank you very much for your appreciation, in light of our results, we have also reflected on this question and completely agree with your comments. In the discussion section, we have included a reflection on the possible association between longer hospital stays and the rates of comorbidities presented by patients upon admission, emphasizing the idea that being hospitalized is a risk factor in itself (lines 286-291).

We are currently designing a new study in which factors such as burden of comorbidities, length of hospital stay, and the medical department in which patients were admitted, among others, will be taken into account when matching the case and control groups. In the meantime, we appreciate your understanding and would like to point out that we have included this aspect in the section on limitations and in the section on further research (lines 330-334 and 338-340).

Subjectivity of "level of dependency" assessment

Thank you for your comment. We have tried to clarify this point in lines 153-155. Although nurses use their clinical judgement to assess the patient's level of autonomy, they perform and record a comprehensive standardized assessment of the patient based on Virginia Henderson's needs. No inter-rater reliability assessment has been performed because this is a standardized practice that nurses carry out in their daily practice, on the basis of which the care plan is developed. We agree with you, while this is a common nursing practice, its subjectivity could introduce variability, so it has been included in the limitations as a potential bias (Line 336-338).

Underdeveloped limitations section

Thank you very much for bringing this point to our attention. We have expanded the limitations section, taking all your suggestions into account (Lines 334-338).

Strength of claim regarding BPG program effectiveness

Thank you for your comment We fully agree with you and we have changed paragraph lines 275-280 according to your suggestion.

Insightful pre-fall length of stay

Thank you for helping us delve deeper into the aspect that the length of stay until falls exceeds the average length of stay during the study period.

The hypothesis that longer length of stay may be associated with a greater burden of baseline comorbidities in patients, certainly makes sense. However, this could also be associated with more complex diseases that required longer diagnosis or treatment times until resolution at discharge. We are currently planning a new study taking these factors into consideration. Therefore, we consider it important to emphasize the idea that being hospitalized is a risk factor in itself, as supported by the study by Seń M. et al. in 2022.